# Identification and Relative Quantification of hFSH Glycoforms in Women’s Sera via MS–PRM-Based Approach

**DOI:** 10.3390/pharmaceutics13060798

**Published:** 2021-05-27

**Authors:** Chiara Melchiorre, Cerina Chhuon, Vincent Jung, Joanna Lipecka, Francesca Di Rella, Alessandro Conforti, Angela Amoresano, Andrea Carpentieri, Ida Chiara Guerrera

**Affiliations:** 1Department Chemical Sciences, University of Naples Federico II, Monte S. Angelo-Cinthia, 80126 Naples, Italy; chiara.melchiorre@unina.it (C.M.); angamor@unina.it (A.A.); acarpent@unina.it (A.C.); 2Proteomics Platform Necker, Université de Paris–Structure Fédérative de Recherche Necker, INSERM US24/CNRS UMS3633, 75015 Paris, France; cerina.chhuon@inserm.fr (C.C.); vincent.jung@inserm.fr (V.J.); joanna.lipecka@inserm.fr (J.L.); 3Department of Senology, Istituto Nazionale Tumori IRCCS “Fondazione G. Pascale”, 80131 Naples, Italy; f.dirella@istitutotumori.na.it; 4Department of Neuroscience, Reproductive Science and Odontostomatology, University of Naples Federico II, 80138 Naples, Italy; alessandro.conforti@unina.it; 5Istituto Nazionale Biostrutture Biosistemi-Consorzio Interuniversitario-Rome, 00136 Roma, Italy

**Keywords:** PRM–mass spectrometry, FSH, glycosylation

## Abstract

Follicle-stimulating hormone (FSH) is a glycohormone synthesized by adenohypophysis, and it stimulates ovulation in women and spermatogenesis in men by binding to its receptor (FSHR). FSHR is involved in several mechanisms to transduce intracellular signals in response to the FSH stimulus. Exogenous FSH is currently used in the clinic for ovarian hyperstimulation during in vitro fertilization in women, and for treatment of infertility caused by gonadotropin deficiency in men. The glycosylation of FSH strongly affects the binding affinity to its receptor, hence significantly influencing the biological activity of the hormone. Therefore, the accurate measurement and characterization of serum hFSH glycoforms will contribute to elucidating the complex mechanism of action by which different glycoforms elicit distinct biological activity. Nowadays ELISA is the official method with which to monitor serum hFSH, but the test is unable to distinguish between the different FSH glycovariants and is therefore unsuitable to study the biological activity of this hormone. This study presents a preliminary alternative strategy for identifying and quantifying serum hFSH glycoforms based on immunopurification assay and mass spectrometry (MS), and parallel reaction monitoring (PRM) analysis. In this study, we provide an MS–PRM data acquisition method for hFSH glycopeptides identification with high specificity and their quantification by extracting the chromatographic traces of selected fragments of glycopeptides. Once set up for all its features, the proposed method could be transferred to the clinic to improve fertility treatments and follow-ups in men and women.

## 1. Introduction

Follicle-stimulating hormone (FSH) belongs to the gonadotropin glycoproteins hormone family, and it has a central and essential role in human reproduction. FSH characterization is essential to elucidating reproductive physiology-regulating fertility, and to improving the diagnosis and treatment disorders of reproduction [1]. FSH consists of two subunits, a 92-amino acid α-subunit and a 111-amino acid β-subunit [2]. The FSH α-subunit is common to all the gonadotropins, and its high degree of structural conservation among different mammalian species implies an important role in the evolutionary process [3]. In contrast, the β-subunit is unique to FSH [2] and binds specifically to its G protein-coupled receptor (FSHR). Hormone binding implies conformational changes of the FSHR that transduce the signal via direct protein interactions at the plasma membrane, resulting in a cascade of biochemical reactions that constitute an intertwined complex signaling network [4,5]. Four naturally human FSH (hFSH) glycovariants have been identified, which differ in their glycosylation [6,7,8]. In all four glycovariants the α-subunit’s *N*-glycan sites (Asn-52 and Asn-78) are always glycosylated, whereas the *N*-glycosylation sites (Asn-7 and Asn-24) occurring within the β-subunit can be (i) both glycosylated, (ii) mixed (one glycosylated) or (iii) non-glycosylated [6,9,10,11]. Although the FSH glycovariants-mediated signaling network is not yet fully deciphered, it is well known that glycosylation at Asn-78 on the α-subunit significantly increases the receptor’s binding affinity; likewise, glycosylation at Asn-52 was found to play an essential role in signal transduction because its removal resulted in significantly decreased potency [12,13]. Overall, the glycosylation level (di, try or tetra-glycosylated FSH) of the hormone has a significant impact on binding and functional assays [14,15]. Furthermore, the wide microheterogeneity of carbohydrates on the FSH N-glycan sites, especially on β-subunit ones, increases the hormone complexity [10], generating a heterogeneous population of different glycoforms, which changes under different physiological and pathological conditions [16,17]. Moreover, recent studies showed that the relative concentrations of these glycoforms vary over the course of the menstrual cycle and the lifetime [11], indicating that these differences in glycosylation may have physiologic relevance [18,19,20]. Nowadays, exogenous FSH is used clinically to induce ovulation in women as part of assisted reproductive technology (ART) treatments (i.e., ovulation induction and controlled ovarian stimulation) and to treat infertility connected to gonadotropin deficiency in men [21]. The accurate measurement and characterization of serum hFSH is essential for patient condition monitoring before and following the treatment. Furthermore, the careful characterization of exogenous FSH injected to the patients is essential for safe and successful treatments [22]. The enzyme-linked immunosorbent assay (ELISA) is currently the reference method for quantifying serum FSH. Although ELISA tests are successfully used in diagnostics for the detection of many proteins, some clinically important targets are still not not reliably detectable, because of the lack of specific antibodies recognizing all the proteoforms generated by post-translational modifications (i.e., FSH), especially in very complex samples such as serum. ELISA-based methods are unable to determine the different FSH glycoforms, a crucial point for the preparation and the characterization of standards and therapeutic products, since the variation of FSH glycosylation, and in particular the degree of terminal sialyation, determines the half-life of the hormone and therefore its in vivo bioactivity [22,23]. An alternative to ELISA is mass spectrometry (MS)-based targeted protein assays, such as MS in multiple reaction monitoring (MS–MRM) or parallel reaction monitoring (MS–PRM). Previous studies have demonstrated the robustness, accuracy and precision, of MS-MRM and PRM analytical methodology for protein quantification [24,25,26,27]. A PRM-based targeted method has been successfully applied in the validation of the relative abundances of proteins and their post-translational modifications (PTMs), such as glycosylation [28,29]. However, the development of reliable analytical MS methods to quantify and characterize the circulating glycoforms of this hormone is a great challenge due to their huge heterogeneity and their very low concentrations in serum. Moreover, MS analysis of glycopeptides is particularly challenging due to their low ionization efficiency and the lack of isotope-labeled synthetic glycopeptides used as standards for absolute quantification [30]. Many strategies are currently used to make glycoprotein MS analysis more effective [31,32,33]. Affinity chromatography with concanavalin A has been applied to the purification and concentration of glycoprotein hormones [34,35,36,37,38]. However, lectin affinity chromatography only allows the separation of non-glycoproteins from glycoproteins; therefore, since half of serum proteins are glycosylated, this approach is not particularly useful for purifying a specific glycoprotein [31]. Immunoaffinity chromatography offers potentially greater specificity, as it has already been successfully applied to purify several non-glycoprotein hormones, and it can allow the enrichment of the population of proteoforms of the target protein [39,40,41,42]. However, for FSH, one of the possible problems using this purification method is the dissociation of hormones into two subunits during elution, resulting in a partial purification of the hormone [43,44]. This study presents an IP enrichment followed by a MS–PRM based strategy to identify and quantify hFSH glycoforms in human serum samples. The proposed strategy based on advanced biomolecular mass spectrometry methodologies, even if still preliminary, could be complementary to immunoassays commonly used for the detection of FSH and other gonadotropins in the clinical setting. The novelty of the PRM-based methods resides in the possibility to discriminate among the different glycoforms of the hormone (thereby expanding information about its functionality) with high specificity and sensibility, overcoming the major limits of ELISA tests. The overall workflow of PRM experiment carried out in this study is summarized in Figure 1.

## 2. Materials and Methods

### 2.1. Immunopurification FSH from Serum

As hFSH glycosylation differs significantly in the different stages of women’s fertility [11], in this study we choose to analyze a serum pool to test the applicability of the method on a range of glycoforms as wide as possible. For that purpose, 54 fertile women’s serum samples (age range: 29 ± 3), supplied by Istituto Nazionale Tumori (INT) “Fondazione G. Pascale” of Naples, were pooled and am hFSH immunopurification protocol was carried out. In total, 500 μL of anti hFSH sepharose CL-4B immunoresin was washed in binding buffer (BB: Tris-HCl 0.1 M, NaCl 0.5 M, NaN_3_ 0.02%). The 500 μL serum pool was diluted 5 times in BB and incubated in batches with immunoresin at 4 °C for 18 h. The immunoresin was washed 5 times with AMBIC 0.1 M and eluted in NH_4_OH 1 M. The eluted fraction was desalted with PD-10 Desalting Columns contain Sephadex G-25, and lyophilized.

Three different aliquots of serum samples were spiked with known amounts of standard FSH in order to be able to calculate IP recovery (details are reported in Appendix A).

### 2.2. Reduction, Alkylation and Enzymatic Digestion

A dried pellet from immunopurification was treated with 100 µL of 8 M guanidine chloride buffer (GuCl)—1 mM EDTA, 130 mM Tris-HCl, pH 7.6. The disulfuric bond reduction was performed by adding 6 µL of 500 mM dithiothreitol under stirring for 60 min at 37 °C. The samples were subsequently alkylated by adding 12 µL of 500 mM iodoacetamide in the dark for 30 min at room temperature. The reduced and alkylated sample was then purified from excess of reagent by CHCl_3_/CH_3_OH/H_2_O precipitation. Enzymatic hydrolysis was carried out in 50 µL of digestion buffer (2 M urea, 50 mM Tris-HCl at pH 8.0) using Chymotrypsin (Roche) with an enzyme:substrate ratio 1:50 at 37 °C for 16 h. As a positive control, 30 g of recombinant standard FSH (RHS_FSH), provided from Merck KGaA, was digested following the same protocol described above. The peptide and glycopeptides mixture from hFSH immunopurified from serum (IP_hFSH) and standard RHS_FSH were resuspended in 10% ACN, 0.1% (*v*/*v*) TFA for LC–MS/MS analysis.

### 2.3. Mass Spectrometry Analysis

One-twentieth of chymotryptic digested samples (IP_hFSH and RHS_FSH) were analyzed using a nano-RSLC-Q Exactive PLUS (Dionex RSLC Ultimate 3000, Thermo Scientific, Waltham, MA, USA). Peptides were separated on a 50 cm reversed-phase liquid chromatographic column (Pepmap C18, Dionex) with a 60 min gradient. Peptides eluting from the column were analyzed by data dependent MS/MS, using the top ten acquisition method. Instrument settings: Resolution was set to 70,000 for MS scans and 17,500 for the MS/MS scans. The MS automatic gain control (AGC) target was set to 3.106 counts with 60 ms for the injection time, and the MS/MS AGC target was set to 1.105 with 120 ms for the injection time. The MS scan range was from 400 to 2000 m/z. Dynamic exclusion was set to 30 s. The mass spectrometry data were analyzed by Proteome Discoverer v2.4 (Thermo Scientific) against the human subset from UniProtKB/Swiss-Prot complete proteome database using the Byonic node search engine (v3.6.0 Protein Metrics, PMI-Suite) with the following settings: Fixed and variable modification: carbamidomethyl (+57.021464@C|fixed), oxidation (+15.994915@M|variable), Gln- > pyro-Glu (−17.026549@NTerm Q|variable); glycan modification: N-glycan 132 human.txt @NGlycan; cleavage residues: FLWY; digest cutter: c-terminal cutter; maximum number of missed cleavages: 2; fragmentation type: QTOF/HCD; precursor tolerance: 5.0 ppm; fragment tolerance: 0.05 Da.

Byonic automatically filters both peptide-spectrum matches (PSMs) and proteins by default; PSMs were filtered by checking the “Automatic score cut” box. The threshold was in the range 200–400. The protein FDR was set at 1% (or 20 reverse count). All proteomic data were deposited on ProteomeXchange Consortium via the PRIDE (Appendix B).

### 2.4. PRM Method

For the PRM method, two scan events were used for each cycle: one full scan followed by a time-scheduled PRM scan. The full scan was acquired at resolution of 35,000 at m/z 200. The AGC target was set to 3 × 10^6^ counts with 60 ms for the injection time. The MS range was from 150 to 2000 m/z. The time-scheduled PRM scans were acquired at a resolution of 17,500 at m/z 200. Other parameters included an AGC target of 2 × 10^5^ counts with 100 ms for the injection time. Fragmentation was performed using a normalized collision energy of 35 eV. The inclusion list containing the precursor ions was used to trigger the acquisition with 5 min scheduled scan windows. To determine the ion’s start and end scan times, we used the data acquired in full scan mode. The cycle time was 2 s and the number of points per chromatographic peak was around ten.

### 2.5. Glycopeptides Enrichment

The remaining part of the IP_hFSH sample was enriched by mixture of lectins using modified Yang, Z. & Hancock W. S. (2004) [45] multi-lectin affinity column protocol. Concanavalina A (ConA), wheat germ agglutinin (WGA) and Ricinus communis agglutinin (RCA), purchased from Sigma, were chosen to enrich all common human FSH N-glycans [9,45]. The sample was resuspended in 40 µL Binding Buffer (BB: 20 mM Tris-HCl pH 7.6, 1 mM MnCl_2_ 1 mM CaCl_2_, 150 mM NaCl) and transferred to a filer unit (Micron YM-30). In total, 36 µL of C.W.R. (15 µL of [ConA] = 6 mg/mL + 15 µL of [WGA] = 6 mg/mL + 6 µL of [RCA] = 18.89 mg/mL) mix was added to the filter unit, mixed at 600 rpm in a thermo-mixer for 1 min and incubated without mixing for 60 min at room temperature. The peptides were washed 4 times by adding 200 µL BB and discarding, which was followed by centrifuging at 14,000× *g* for 10 min. Glycopeptides were eluted in 200 µL elution buffer (EB: 20 mM Tris, 0.5 M NaCl, 0.17 M methyl-α-d-mannopyranoside, 0.17 M N-acetyl-glucosamine and 0.27 M galactose, PH = 7.4). The buffer of eluted fraction containing glycopeptides was then exchanged against water and concentrated using Amicon Ultracel 3k centrifugal filters. One-thirtieth of eluted enriched glycopeptides were injected in the mass spectrometer nanoRSLC-Q Exactive PLUS (Dionex RSLC Ultimate 3000, Thermo Scientific, Waltham, MA, USA) as described before and analyzed using PRM modality.

## 3. Results and Discussion

### 3.1. Recombinant FSH Full Scan LC–MS/MS Analysis: PRM Method Development

In order to develop the targeted data acquisition methods (MS–PRM), we have used preliminary discovery data acquisition from shotgun experiments of recombinant standard human (RSH) FSH. For this purpose, key information about the target glycopeptides (m/z of the precursor ions, charge state and start-end elution time) were obtained from classical LC–MS/MS full scan analysis of RHS_FSH chymotryptic digestion. Protein identification was performed by the Byonic search engine, and each recognized glycopeptide was then validated by manually interpreting MS/MS spectra. As an example, the identification of a glycopeptide (m/z = 1253.30) derived from α-FSH is shown in Figure 2. The full scan LC–MS/MS TIC chromatogram is reported in Figure 2A. The signal at m/z 1253.30, found in the full MS spectrum at 20.28 min (Figure 2B), was assigned to RSKKTMLVQKN_52_VTSESTCCVAKSY peptide with complex biantennary fully sialylated glycosylation at Asn-52 of the α-subunit. We obtained an excellent HCD MS2 fragmentation spectrum containing signals attributable to both the peptide and the glycan moiety, thereby confirming the assigned modified peptide (Figure 2C).

Previous reports suggest that fragmentation in ETD mode is necessary for informative fragmentation of glycopeptides [46,47]. Our results provide precious evidence of glycopeptides characterization using HCD fragmentation. In the same way we characterized the multiple glycan moieties of three N-glycan sites: α-subunit Asn-52–78, and Asn-24 of β-subunit. Unfortunately, no glycopeptides on the other β-subunit *N*-glycan site (Asn-7) were identified by Byonic software, probably due to the greater number of branched structures on the β-subunit at Asn-7 compared with those at Asn-24. In fact, highly branched tetra-antennary glycan structures have only been reported at Asn-7 [8,11]. This result suggests higher signal suppression of such highly sialylated structures on Asn-7, making their identification extremely complex. However, manual spectra interpretation led to the identification of four possible signals attributable to the peptide L.TN_7_ITIAIEKEECRF.C with four different glycovariants. Due to the very low ionization efficiency of these ions, no fragmentation spectra could be detected; therefore, the identification of the four glycoforms on Asn-7 could be performed on the basis of the exclusive molecular weight of the entire glycopeptide and of the retention time relative to the unmodified peptide (see Appendix A for Full LC-MS/MS identification of RHS_FSH standard). These results are a key step in our workflow, as they constitute a library of retention time, and MS and MS2 spectra of target glycopeptides (see Glycopeptide’s library in Appendix A) were included into the target list for the development of the MS–PRM method to identify serum hFSH glycopeptides (see Figure 1 for a schematic overview).

### 3.2. Characterization of hFSH Purified from Serum Using Full Scan LC–MS/MS Analysis

Human serum is a very complex matrix due to its high protein content (60–80 mg/mL) and the extremely dynamic concentration ranges for those proteins. The glycoproteome is a major subproteome present in human serum; in fact, about 50% of all serum proteins are glycosylated (about 150 detectable glycoproteins) [48]. The glycoprotein hormone FSH is present at very low levels in serum, as in fertile women its average level in cycle phases is 5.8 mIU/mL (~0.39 ng/mL) [10]. To detect and characterize this glycoprotein in such a complex biological matrix, purification and enrichment steps before MS analysis are essential [33]. Anti-hFSH sepharose CL-4B Immunoaffinity purification was carried out on three aliquots of pooled serum samples as described in Materials and Methods. Although the immunopurified sample identification showed significant copresence with other human serum proteins (Appendix A), the applied strategy has proven to be successful in purifying and identifying both α and β-subunits of serum hFSH with 18% and 34% sequence coverage, respectively (Figure 3). The low percentages of the FSH sequence coverage were due to the wide heterogeneity of glycovariants and the presence of other, more abundant serum proteins, whose signals can hinder the detection of hFSH glycopeptides. Nonetheless, we could identify one glycopeptide, the complex biantennary glycan structure on site Asn-24 of β-subunit (Figure 3).

The presence of the unmodified peptide CISINTTW confirmed the partial glycosylation of β-FSH [14]. The extracted-ion chromatogram (XIC) relative to the glycopeptide CISIN*TTW m/z = 1115.77 (RT: 49.01 min) compared to the unmodified one m/z = 994.46 (RT: 48.48 min) shows that the intensity of the latter was approximately 10-fold higher than that of the glycosylated one (Appendix A). Since glycosylated peptides have lower ionization efficiency than the relative unmodified ones, MS signals of contaminant peptides could hide the IP_hFSH glycopeptides, confirming the above-mentioned hypothesis.

Additionally, the recovery of the purification was calculated (see Appendix A Immunoprecipitation recovery). In order to detect FSH even in not-depleted serum (input), three new aliquots of pooled sera were spiked with a known amount of FSH before immunoprecipitation. The input, the flow through (FT) and the immunoprocipitated (IP) spiked serum samples were analyzed by PRM developed method. No peptides belonging to FSH were recorded in FT samples. Recovery value (89%) was calculated on the Quantifier (best transition: b10-1107.5364) by comparing the average area of extracted peptide ETVRVPGCAHHADSLY (906.4309 m/z, z = 2) yield before and after the immuprecipitation (Appendix A). The reproducibility of the IP was calculated by measuring the area of peptide ETVRVPGCAHHADSLY representative of the amount of hFSH in each experiment (see Appendix A for details).

### 3.3. Detection of hFSH Glycoforms Purified from Serum by MS–PRM Analysis

To improve the glycopeptides’ identification and to obtain their relative quantifications, the same aliquot of serum IP_hFSH was re-analyzed with MS–PRM target method. We applied the MS–PRM method developed on standard RHS_FSH to the analysis of hFSH purified from serum. Our method included all FSH N-glycan sites identified (4) in 24 different glycoforms (Figure 4). MS–PRM analysis improved the identification of the first untargeted LC–MS/MS glycopeptides from the serum sample because when analyzing complex peptides mixtures, the data-dependent acquisition method suffers from a bias toward sampling and identification of most abundant peptides, resulting in poor and inconsistent detection of low-level glycopeptides.

In addition, PRM allows an in-depth analysis of all potential glycopeptides present in the sample because only one predetermined charge state of a given glycopeptide will be selected for fragmentation, allowing us to identify, in IP_hFSH serum sample, the great microheterogeneity of glycosylation, which characterizes the human follicle stimulating hormone (Appendix A). This method allowed us to identify nine glycopeptides instead of only one obtained with the untargeted analysis, proving the great efficiency of the MS–PRM method.

### 3.4. Glycopeptide Enrichment Prior PRM Analysis

To investigate the presence of unidentified target glycopeptides in serum, the IP_hFSH sample was subjected to glycopeptide enrichment and a further MS–PRM experiment. Enriched PRM data analysis of glycopeptides allowed the identification of four additional glycoforms that could not be identified with the previous MS–PRM experiment. Figure 4 reports the comparison of glycopeptides glycopeptide in all experiments on the IP_hFSH serum sample (full scan LC–MS/MS and MS–PRM, MS–PRM on the enriched sample) and the standard RHS_FSH target glycopeptides (see Appendix A for details). Although glycopeptide enrichment allows the identification of additional minority glycoforms, this analysis has the disadvantage of a further purification step which negatively affects the quantitative analysis. Therefore, we processed the data obtained from the MS–PRM analysis performed on the non-enriched IP_hFSH sample for the subsequent quantitative analysis.

### 3.5. Comparison of the Glycoprofiles of RHS_FSH and hFSH from Serum

The preliminary building of a reference library containing target glycopeptide MS2 spectra played a central role in the workflow as both data acquisition and data processing. The MS–PRM data acquisition method has the advantage of being relatively easy to build because a priori selection of target transitions is not required, as in MS-MRM. The absolute quantification of serum hFSH glycopeptides is extremely difficult, as there is no availability of isotope-labeled standard glycopeptides on the market. However, a relative quantification of hFSH glycosylation can be obtained thanks to the high specificity of MS–PRM. The high resolution allows the chromatographic traces to be extracted using tight mass tolerances, which improves the differentiation of analyte signals from co-eluting interferences, resulting in increased confidence and better analytical performance. The quantification was only applied to the glycopeptides with identities confirmed by MS2 spectrum (Asn-52–78 of α-subunit and Asn-24 of β-subunit), and was performed by extracting, post-acquisition, the chromatographic traces of specific fragments using Skyline v20.1.0.76 software. The extracted ion currents (XIC) associated with the different transitions (precursor-ion → fragment-ion) for the same glycopeptide were recorded at the same retention time, and for each glycopeptide, the most intense transition was selected as the quantifier (Appendix A). This finding ensured unequivocal identification and allowed us to disclose the relative distributions of glycans on each N-glycan site except Asn-7, as no MS2 spectra were recorded at this site (Figure 5). As a proof of concept of the utility of the reported method, we compared RHS_FSH with hFSH from a pool of woman’s sera. Figure 5 shows the MS–PRM relative quantification results expressed as relative percentage distributions of glycan moieties on the glycosylation sites of both recombinant and serum-purified hormones. Some differences could be appreciated. In IP_hFSH we could observe the absence of high mannose (Man6) glycan structures on α-FSH Asn-52 and β-FSH Asn-24, and there was a similar high abundance of complex glycans which were largely coated with sialic acid. In both samples we found on Asn-52 the predominance of the mono-sialylated complex biantennary structure (A2G2S1), and on Asn-78 the most abundant structure was the completely sialylated one (A2G2S2). Asn-24 of the β-subunit showed more heterogeneity in glycan structures. Comparing the glycan structure abundance with serum IP_hFSH, recombinant protein exhibited higher number/level of biantennary glycans, most of them with core-fucosylation with respect to the serum FSH. This observation agrees with a previous study, which also suggested an additional feature of recombinant FSH oligosaccharide structures—the presence of antenna-fucosylation, which is rare in serum hFSH [14,49].

## 4. Conclusions

Glycopeptide analysis is a very challenging task that requires expertise in sample preparation, high sensitivity and specificity, an analytical method and dedicated software to analyze data. Thanks to our approach we could obtain preliminary but fine glycan characterization of the FSH glycohormone (involved in human fertility)—even including the site-specificity. To the best of our knowledge, we provided for the first time a MS–PRM-based strategy for the characterization of an hFSH glycan moiety from serum and its site-specific relative quantification. As a whole, our data lay the groundwork for developing a quantitative analysis addressed toward glycopeptides, which is not possible with the standard ELISA assays. Once optimized, the proposed strategy could be of great interest for understanding hFSH glycosylation’s involvement in the complex intracellular signaling mechanism that impacts the human reproductive system. The identification of different hFSH glycoforms and the elucidation of their functional roles could be two steps forward in delineating the clinical relevance of some glycoforms over others, with the ultimate goal of developing or improving drugs for the treatment of infertility. Another advantage of this strategy is its ability to detect several analytes simultaneously without affecting the specificity and sensitivity [50], unlike clinical multiplexed ELISA. For this reason, we believe that this method could be expanded and implemented for monitoring all the gonadotropins in a single analysis. Furthermore, quantification could be enhanced by measuring the relative glycopeptide/peptide ratio by inserting the detection of hormone-specific peptides into the PRM method. This could bring estimates of the glycans and stoichiometry of the whole glycoprotein. Given the numerous advantages of mass spectrometry in PRM mode, we believe that this method, in time, could replace ELISA tests as a reference standard for the detection of FSH in the clinical setting.

## Figures and Tables

**Figure 1 pharmaceutics-13-00798-f001:**
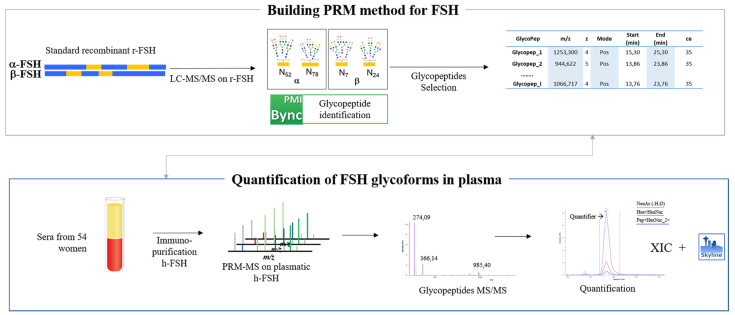
Workflow of the PRM experiment carried out to identify and quantify h-FSH glycoforms in the serum sample.

**Figure 2 pharmaceutics-13-00798-f002:**
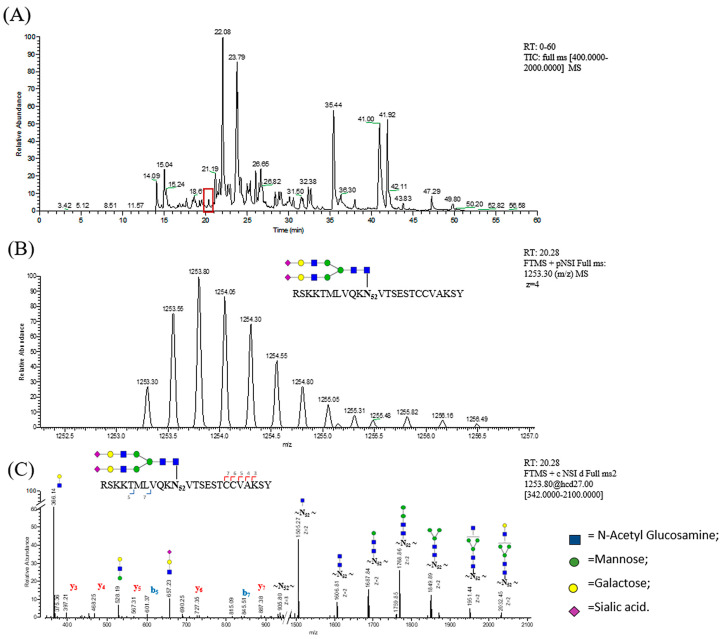
Characterization of RHS_FSH α-chain glycopeptide: RSKKTMLVQKN_52_VTSESTCCVAKSY. (**A**) LC/MS full MS TIC chromatogram, in the red box is highlighted the peak at RT 20.28; (**B**) full mass scan at RT: 20.28 and 1252.5–1257.0 m/z zoom; (**C**) MS2 spectrum of the glycopeptide ion is consistent with the proposed glycan structure.

**Figure 3 pharmaceutics-13-00798-f003:**
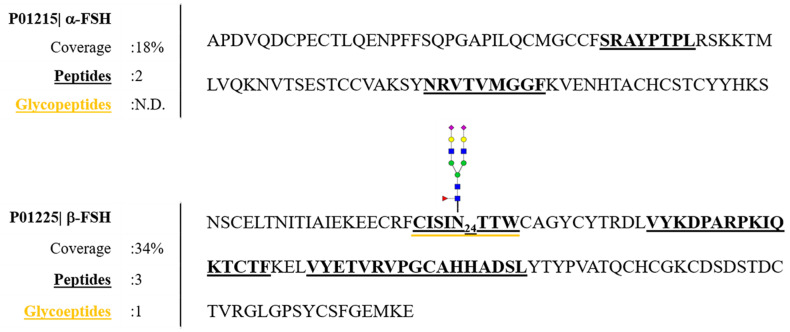
Full scan LC–MS/MS allowed the identification of both α and β-subunits of hFSH in the serum sample. One glycopeptide (glycan composition: Fuc(1)HexNac(4)Hex(5)NeuAc(2)) on β-FSH has been recognized.

**Figure 4 pharmaceutics-13-00798-f004:**
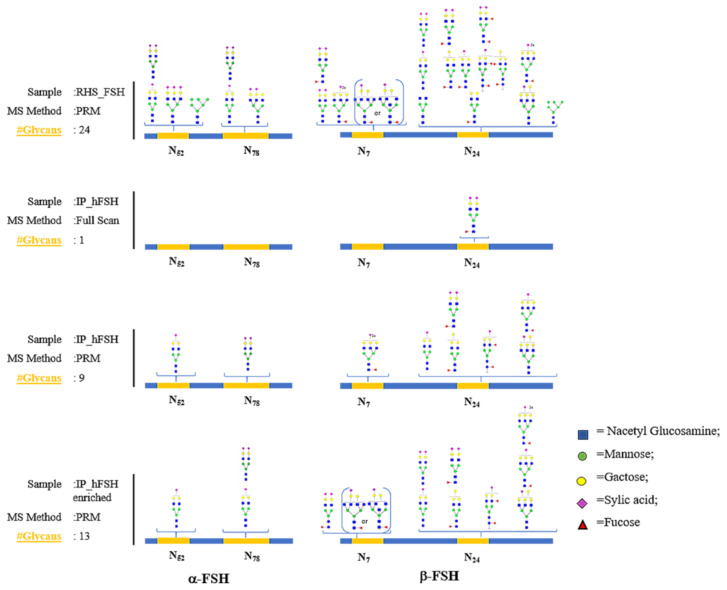
Comparative descriptions of the glycans present on each of the glycosylation sites in standard RHS_FSH and plasmatic IP_hFSH in all experiments: Full scan LC–MS/MS and MS–PRM of IP_hFSH and MS–PRM of IP_hFSH_Glycoenrich. In the figure are only the glycopeptides identified according with defined quality criteria described in Materials and Methods.

**Figure 5 pharmaceutics-13-00798-f005:**
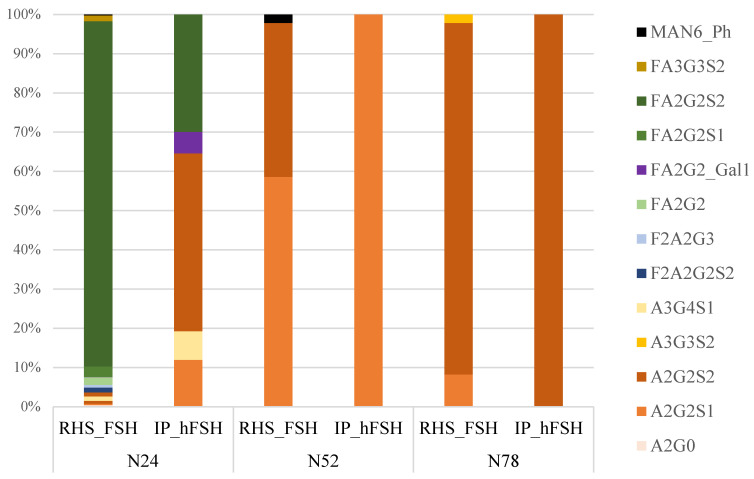
MS–PRM relative quantification results expressed as relative percentage distributions of glycan moieties on the glycosylation sites of both recombinant (RHS_FSH) and plasmatic (IP_hFSH) hormones.

## Data Availability

Data available in a publicly accessible repository. The data presented in this study are openly available in [repository name PRIDE] at link https://www.ebi.ac.uk/pride/.

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
