# Peer review of "Identification and Relative Quantification of hFSH Glycoforms in Women’s Sera via MS–PRM-Based Approach"

_pharmaceutics, 2021, doi:10.3390/pharmaceutics13060798_

Round 1
Reviewer 1 Report
After revision and new experiments described, the article warrants now publication in Pharmaceutics.
Author Response
We thank reviewer 1 for the positive comment.
Reviewer 2 Report
The article deals with the attempt to quantitate the glycoforms of human FSH. The topic is interesting and the approach promising. The experiments are properly performed and the conclusions are pertinent. The text requires to be improved. Punctuation and grammar are sometime misleading. Below there are some examples.
Line 22: is will
Line 49: while,
Line 222: high proteins content
Line 229: three aliquot
246: contaminant proteins peptides MS signals
252: No peptides belonging FSH
And so on.
Line 352 Glycopeptides Library, not found
Reviewer 3 Report
The manuscript „Identification and relative quantification of hFSH 2 glycoforms in women sera by PRM MS based 3 approach” describes the setup of a PRM method starting with the identification of glycopeptides. The characteristics of these peptides are then used to quantify serum hFSH glycoforms, which is not possible with the standard ELISA assays.
comment:
Quantification of low abundant peptides in a complex matrix is a challenging task. It becomes much more complicated if glycopeptides should be quantified. The authors use PRM which is an established method for peptide quantification and there are interesting results provided by the authors. The data are, however, very preliminary, because for the identification of the peptides of interest, no quality criteria are provided. In addition, quality criteria of reproducibility should be provided on the basis of intensities of peptides of interest. Taken together it is not clear to me whether the method is applicable for the quantification of glycoforms of FSH in its current state.
A crucial point of MRM or PRM assays is the characterization of the target peptide with precursor mass, suitable fragment mass and retention time, which is easy for unmodified peptides as synthetic peptides are routinely used. Because that is not possible for most glycopeptides it must be convincingly shown that the signals observed in PRM are caused by the proposed glycopeptides. In this case shotgun proteomics and the Byonic software packages was used to do that.
- Many of the fragment spectra shown in glycopeptides library are of low quality. Whereas in shotgun proteomics a single fragment spectrum is used to get a peptide sequence, many MS2 spectra of a peptide across the chromatographic peak are acquired. Why not using an averaged MS2 spectrum of a PRM analysis to get higher quality sequencing data.
- Because the identification of the glycopeptides is such crucial a MS2 spectrum of each glycopeptide must be accessible for the readers. A good example (figure 2) is presented, but that is not sufficient. In figure 2 the fragmentation of the glycan tree is observed with high signal intensities, but the fragments for peptide sequencing (b- and y-ions) are close to the background, which is a problem of HCD.
- The Byonic software was used to analyze the data. Whereas the settings of the software are described in the method part, there is no threshold or significance level given, above which the identification is reliable. In the glycopeptides library table, scoring values are given, but again, there is no hint above which score a result is significant. Only those glycopeptides which are identified according defined quality criteria should be mentioned in the text. If there are arguments that manuell inspection of a fragment spectrum leads to a reliable identification even when the statistics don't meet the threshold, then these arguments should be given in the glycopeptides library table.
- In a PRM method, precursor ions are selected by a window of about 1 Da (this value should be included in the method part) and fragments are measured with high mass accuracy. For the presented PRM method it means the precursors consisting of a peptide and a glycan moiety are selected with low mass accuracy. The fragments measured with high mass accuracy are however common to many glycopeptides as m/z= 366.14 for HexNAcHex. Quantification would be more significant, if a fragment would be used which contains the peptide part like m/z=1505.27 in the first fragment spectrum of the glycopeptides library.
By shotgun proteomics analysis of hFSH purified from serum a glycopeptide was identified as stated in the text, but no fragment spectrum is shown in the supplement that the peptide eluting at 49.55 min (figure S1) is indeed the proposed peptide.
The recovery of the purification was calculated (89%). It is not clear how it was calculated. From the intensities observed in figure S2, I would conclude that there is 40 times more peptide in IP_spike compared to input_spike.
line 256-257: Reproducibility should be shown by providing the intensities of the detected FSH peptides.
Figure 4: only such glycopeptides should be shown which are identified according defined quality criteria.
Round 2
Reviewer 3 Report
The manuscript „Identification and relative quantification of hFSH 2 glycoforms in women sera by PRM MS based 3 approach” describes the setup of a PRM method starting with the identification of glycopeptides. The characteristics of these peptides are then used to quantify serum hFSH glycoforms, which is not possible with the standard ELISA assays.
I had two major concerns: A major concern from my side was whether the identification of the listed glycopeptides is reliable. This point is clarified. The authors pointed to a publication which show that the identification of the glycopeptides was done in a scientifically sound way. The other area of questions was about reproducibility and quality control of the method. The answers were not really satisfying. This is still a part of the manuscript which must be improved in my opinion.
Comment to response 1:
- Detection of a similar number of common contaminants is not an appropriate quality control for a specific immunoprecipitation. Therefore, figure S3 is misleading and should be removed.
- Calculation of the ratio between area and height of a signal is not an appropriate way to estimate the reproducibility of a method. IP is done to enrich for hFSH2. Reproducibility of the method can be quantified by measuring the amount of hFSH2 in each experiment, which correlates with the intensity of peptide ETVRVPGCAHHADSLY. This is an area of 76709242 in experiment two (…ip_spike_2_10220) compared to 8494184 in experiment three (…ip_spike_3_10221). There is a nine-fold difference between these two experiments. That must be discussed.
Comment to response 2 and 3:
Thanks to the authors for clarifying.
Comment to response 4:
I do not agree with the authors opinion that the probability of co-fragmentation of other glycopeptides is fairly low. Even after enrichment there may be thousands of glycopeptides present in amounts, which do not allow for successful identification, but which contribute to the signal in a PRM method. To use a low resolution selection window as one criteria and a low specific signal as the second criteria is risky.
Comment to response 5:
Thanks to the authors for adding this high quality fragment spectrum
Comment to response 6:
In the method part (line 136) it is written: 1/20 of chymotryptic digested samples (IP_hFSH and RHS_FSH) were analysed using a nano-RSLC-QExactive PLUS …
In line 169: The remaining part of the IP_hFSH sample was enriched ….
Line 182/183: 1/30 of eluted enriched glycopeptides was injected……
Taken from figure S1 it is a peak height for the b10 fragment of about 15E3 counts in A (input) compared to 750E3 in B (spike in). There is a huge difference between the method part and the figure S1 on the one hand and the statement, that the recovery is 89%.
From my calculation I expect a 9E3 signal height in B, if a 15E3 input height is detected in A and 89% recovery is considered.
This difference must be clarified.
Comment to response 7:
Thanks to the authors for adding the table to S3. Because the reproducibility is not high, reasons for that must be discussed.
Comments to response 8:
Thanks to the authors for clarifying.
Round 3
Reviewer 3 Report
The authors responds are ok.
Author Response
We thanks Reviewer 3 for positive comment.
This manuscript is a resubmission of an earlier submission. The following is a list of the peer review reports and author responses from that submission.
Round 1
Reviewer 1 Report
All right. With U being now clearly defined I now endorse publication of this article.
Reviewer 2 Report
The paper deals with a workflow for the accurate measurement and characterization of serum hFSH glycoforms (gonadotropin glycoproteins).The precise description of the glycosylation of hFSH s therefore essential for an optimal treatment for woman fertility.
The strategy adopted is based on mass spectrometry MS/MS using parallel reaction monitoring (PRM) after immunoprecipitation on a pooled serum sample, to allow a wide panel of glycoform to be detected. The workflow has been developed and applied firstly on a recombinant standard human FSH. HCD is claimed to give good MS/MS spectra on the glycopeptides. Manual spectra interpretation was required for precise assignment of peptides glycoforms but what are the attribution criteria? . The retention time were used for further identification.
In real complex matrice, using full full scan spectra, the sequence coverage is very low.
The PRM method improves the identification of different glycoforms.
Comments
The description of the methods is correct and the sequence of steps in the strategy is clear.
The paper is however quite classical in its timeline. The same applies for all proteomics/glycoproteomics methods.The novelty should be better emphazised.
On the negative side, there is a total lack of quality assessment of the data, no repeatability test, no identification criteria based on MS, with is important at the detected signal level.
The strategy has been tested on a pool of serum samples. The paper, resulting from a cautious sample preparation effort and delicate MS development is therefore very preliminary and should be completed by a minimal significant number of individual measurements before claiming for a routine multiplexed test method. In the absence of those, what can be really deduced from fig. 5?
Reviewer 3 Report
This paper describes data on the LC-MSMS analyisis of the Follicle-Stimulating Hormone, isolated from human serum by immunopurification, and after glycopeptide enrichment. The authors describe the identification and characterization of several glycans present at the four glycosylation sites in alpha and beta chains of FSH. Glycan identification improves whem a targeted PRM approach is used for the analysis, based on glycans characterizaded from recombinant hGSH and literature data. An additional improvement is attained after an additional glycopeptide lectin-based enrichment step. The authors claim that the PRM method described can be useful to monitor different glycoforms, of clinical relevance, in serum, while the ELISA merthod of reference is unable to distinguish between glycoforms.
In my opinion, while the authors provide some data that can be of interest in the development of an LC-MS based method for the analysis of FSH, the results are too preliminar at this point to rise any conclusion on the feasibility of such a method. Some of the points that would have to be addressed are the following:
- As the authors point out, given the low levels of FSH present in serum, at least one enrichment step is required to reach enough sensitivty. From the results shown of the paper, even a second enrichment at glycopeptide level seem to be unavoidable. A control of the recovery and reproducibility of these enrichment steps would be neccesary. The yield of the immunopurification step colud be assessed for instance by ELISA or WB analysis of serum samples before and after immunopurification. Alternatively, a targeted MPRM MS method could be used to monitor a non-glycosylated peptide representative of the overall concentration of FSH.
- Reproducibility of the immunopurification, enzymatic digestion, and glycopeptide enrichment steps should be assessed by replicate analysis of aliiquots of a serum sample.
- In the absence of internal standards for the glycopeptides measured in the PRM analysis presented, the relative quantitation presented in figure 5, can be very biased by the differences in ionisation efficency of the different glycopeptides. The only quantitative comparison possible at this point of method development would be the comparison between different samples.
In order to demonstrate the potential clinical utility of the proposed FSH glycoform analysis method, the authors should present the results of the analysis of individual serum samples (not just a pool of samples), to show that different patterns of glycosylation can be revealed by the method.